# Acute Acoustic Trauma after Exposure to Assault Rifle Noise among Conscripts in the Finnish Defence Forces—A Population-Based Survey

**DOI:** 10.3390/ijerph20043366

**Published:** 2023-02-14

**Authors:** Markku Toivonen, Rauno Pääkkönen, Riina Niemensivu, Antti Aarnisalo, Antti A. Mäkitie

**Affiliations:** 1Department of Otorhinolaryngology—Head and Neck Surgery, University of Helsinki and Helsinki University Hospital, FI-00029 HUS Helsinki, Finland; 2Tmi Rauno Pääkkönen, FI-33720 Tampere, Finland; 3Research Program in Systems Oncology, Faculty of Medicine, University of Helsinki, FI-00014 Helsinki, Finland

**Keywords:** hearing loss, tinnitus, military, weapon, small caliber

## Abstract

Conscripts are exposed to various sources of impulse noise despite hearing protection recommendations. The aim of this study was to investigate the frequency of acute acoustic trauma (AAT) among conscripts after exposure to assault rifle noise in the Finnish Defence Forces (FDF). This nationwide population-based cohort comprised all conscripts (>220,000) in the FDF during the years 1997–2003 and 2008–2010. We included those who claimed to have AAT symptoms from assault rifle noise during the study periods. During the investigated 10 years, 1617 conscripts (annual variation, 75–276) experienced a new hearing loss due to AAT. Altogether, 1456 (90%) of all AAT-induced hearing losses were caused by rifle-caliber weapons and 1304 (90%) of them when firing a blank cartridge. There was no clear diminishing trend in the annual numbers of AATs. In 1277 (88%) incidents, no hearing protector was used. Tinnitus was the most prominent symptom. Hearing losses after AAT were typically mild, but serious deficits also occurred. In conclusion, we found that 0.7–1.5% of the conscripts experienced an AAT during their service in the FDF. Most incidents occurred when firing a blank cartridge with a rifle-caliber weapon and with no hearing protector in use.

## 1. Introduction

Finland has a general law of conscription. Military (or non-military) service is obligatory for every Finnish male citizen between 18 to 60 years of age. Voluntary military service was made possible for women in 1995. Conscript service is usually carried out at the age of 19 years, and it lasts for 165, 255 or 347 days. During the service, a conscript may be exposed to impulse noise from various firearms as the shooting conditions vary both in the field circumstances and at a shooting range [1,2,3,4]. Various types of hearing protectors (earmuffs, earplugs) are available for all exercises, drills, and shooting range activities at the Finnish Defence Forces (FDF), and their protection capacity is reportedly good [4,5,6,7].

There are many possibilities for the exposure to impulse-noise in military environment, including various small arms [5]. Measurements of acoustical energy of shots from rifle-caliber weapons have shown that the peak pressure levels within the shooter’s ear vary from 156 to 170 dB (rifles, pistols) [1]. In these measurements, the (A-) duration was around 0.3 ms. The attenuation (insertion loss) of impulse noise measured with an ear-canal-placed miniature microphone varied from 16 to 23 dB for earplugs, 10 to 20 dB for earmuffs, and 24 to 34 dB for the combined use of plugs and muffs [5,6,7,8,9]. If the limit for the C-weighted peak level is 135–137 dB for unprotected ears or 140 dB for protected ears (EU directive against noise-induced hearing loss), then protection against low-frequency noise is provided for up to 156 dB by earplugs, up to 150 dB by earmuffs, and up to 165 dB by combined use of plugs and muffs [8].

The hearing protection guidelines in the FDF date back to 1968, when hearing protectors (ear plugs) were instructed to be used at shooting ranges. Earmuffs became mandatory a decade later (FDF 1979) for sheltered shooting-range conditions. However, the use of ear plugs only was allowed when shooting at battle shooting situations or at open shooting ranges. According to these regulations, the used hearing protection may not risk general safety or cause undue harm in military operations. It remains obvious that these limitations and the discomfort of various types of ear plugs led to insufficient hearing protection among soldiers. Therefore, the supplementary and revised regulations (FDF 1985) ordered the use of at least earplug protectors for all shooting situations. Further, later regulations (FDF 1989, FDF 1992) gave instructions regarding protector types and safety distances for persons responsible for military education and training.

A strong acoustic trauma will induce histological changes in the cochlea and auditory nerve. Cochlear outer hair cell loss and damage of inner hair cell stereocilia and axonal loss and myelin sheath disorganization of the auditory nerve will consequently lead to permanent hearing loss [10].

Typically, hearing loss after acute acoustic trauma (AAT) can first be seen at 6 kHz frequency, and then at 8 kHz and 4 kHz [9,11]. High frequency AAT-induced hearing loss is most typical, but also flat or low-frequency losses can be seen [9,11]. Tinnitus is the most common symptom related to AAT. In most cases it is even more disturbing than the hearing loss itself [11,12,13,14]. Hyperacusis, distorted sounds or pain in the ear are symptoms that can be present at early phase of cases with AAT [13,14].

There are no reports on the occurrence of AAT in the FDF since the study by Mrena et al. (2004) including a selective series of 163 AAT patients treated at the Central Military Hospital during the year 2000 [14]. Most of the AATs (87.5%) in their study occurred in patients with unprotected ears, the assault rifle being the typical causative weapon [14]. Of note, prior to military service all Finnish recruits are examined with a screening test for their hearing. The quality of these hearing screening tests has proved to be of good quality [15].

The primary aim of the present study was to investigate the population-based occurrence of AATs after exposure to assault rifle noise during conscript training in the FDF. The secondary aim was to evaluate the possible causes for such accidents.

## 2. Materials and Methods

In the FDF a conscript is instructed to immediately report a suspected AAT and fill in a questionnaire (Appendix A), which starts a process that includes hearing screening tests and an evaluation of further actions in the healthcare system. A selected group of conscripts will annually get monetary compensation for a consequent and verified hearing loss. These incidents are also included in the annual reporting at military unit and national level.

This retrospective nationwide population-based study consists of two cohorts covering the years 1997–2003 and 2008–2010. Information on 2004–2007 does not exist, based on organizational causes. The large population consists of all FDF conscripts in service during these two time periods (>220,000). The inclusion criterium was that if the conscript complained of changed hearing after exposure to assault rifle noise, then a questionnaire was filled in and later an audiometric analysis was performed. All conscripts who had filled in the questionnaire were thus included in this study. It was still possible that the audiometric analysis later showed only a rather limited hearing threshold shift. The study cohort comprised all conscripts (1456) during the same time periods who claimed to have symptoms of an AAT, i.e., ear pain, dullness, tinnitus or a subjective hearing deficit after exposure to assault rifle noise. These subjective symptoms, combined with a hearing threshold change of at least 10 dB for one frequency in the speech area (frequencies 500 Hz–4 kHz) or a change of at least 15 dB for the other frequencies, compared with a prior audiogram, were considered an AAT-induced hearing deficit.

The subjects were asked to fill in a questionnaire including the following parameters: type of weapon and cartridge, user of the weapon (own weapon or belonging to another shooter), distance from the weapon, type of hearing protection used, reason for not wearing hearing protection, prior instruction on the use of hearing protection. A written description of the incident was also obtained (Appendix A). These reporting forms were signed by the conscript, the trainer and the healthcare provider.

All Finnish conscripts are screened for their hearing at the beginning of their military service. In this screening examination, a 20 dB hearing level is measured at frequencies of 0.5, 1, 2, 3, 4, 6 and 8 kHz in a standardized auditory booth. If the hearing threshold is higher than 20 dB, then the hearing threshold of this frequency is examined more carefully. After a noise incident, accurate hearing thresholds are always examined.

In cases of an AAT-induced hearing loss, it is compulsory to consult an ENT (ear-nose-throat) specialist for evaluation of further care. If the AAT-related deficit exceeds 10 dB at 0.5–2 kHz or more than 35 dB at 3–8 kHz, then the conscript is ordered to receive hyperbaric oxygen treatment. Currently, corticosteroids and hyperbaric oxygen therapy are the standard treatment of acoustic trauma in the FDF. However, during the study periods, normobaric or hyperbaric oxygen only was used. Milder deficits are monitored with a control audiogram after two weeks. In addition, the conscript is ordered not to attend noise-causing service for a certain period of time.

Sound pressure levels (SPLs) of weapons can be measured by using sound level meters, hearing loss dB is measured by using audiometers, and attenuation of hearing protectors by using an insertion loss factor in field conditions with sound-level meters (microphone outside and inside hearing protector). However, the information data on hearing protector attenuation values are measured in laboratory conditions by using the human threshold of hearing (dB).

The study proposal received approval from the Research Ethics Committee of the Helsinki and Uusimaa Hospital District Occupational Health Ethics Committee (Dnro 315/E2/03/27.6.2003) and from the Coordinating Ethics Committee of the Helsinki and Uusimaa Hospital District (Dnro 41/13/03/00/2012). Study permissions were granted by FDF on 7 May 2003 (Dnro R680/8/E/II) and 29 March 2012 (Dnro 2148/26/2012/AI6100).

## 3. Results

### 3.1. AATs

During the investigated periods (total 10 years) the total number of conscripts with a new hearing loss due to AAT in the FDF was 1617 (annual variation of cases of HL from rifle-caliber noise, 67–258). There were 161 AATs after exposure to noise from large-caliber weapons and explosions. Altogether, 1456 (90%) of all hearing losses were caused by rifle-caliber weapons. This is around 0.7–1.5% of the annual number of conscripts (mean annual number of forms filled in 145.6, health information 428–438, and annual variation, 19,000–30,000). Altogether, 1150 rifle-caliber AATs were registered during the first study period (1997–2003, Group I) and 306 during the second period (2008–2010, Group II). The audiograms of Group I were measured on the same day of the incident in 73%, on the following day in 15%, later in 3%, and the timepoint was unknown in 9%. The audiograms of Group II were measured during the same day in 44%, the following day in 17 %, later in 4%, and on an unknown day in 35%. There were 1439 (99%) males in the study cohort and 17 females (1%).

There were 841 (73% of the Group I) unexpected shots in the Group I and 230 (75%) in the Group II. Normal cartridges caused 16 (1%) AATs in the Group I and 5 (2%) AATs in the Group II.

The annual numbers of AATs from assault rifle-caliber weapons in the FDF showed a slightly diminishing trend during the study periods (Figure 1).

### 3.2. Weapons

Rifle-caliber weapons consisted of assault rifles, pistols and light machine guns. Acute acoustic traumas were mostly noted in February and November, which relates to the first shooting practices that the conscripts perform in their service. Most hearing losses were due to firing an assault rifle with a blank cartridge during military drills (1286, 88% of all 1456 AATs caused by rifle-caliber weapons). When using normal cartridges, there was a recorded AAT-related hearing loss information only in 1% of the cases, typically at shooting ranges. Typically, in 593 (41%) out of the 1456 cases the neighboring conscript was shooting. Among the whole cohort of 1456 persons, the distance was less than 2 m for 973 (67%), 2–5 m for 257 (18%), more than 5 m for 84 (6%), and unknown for 142 (10%).

### 3.3. Hearing Protectors

In 1266 (87%) out of the 1456 cases having completed the questionnaire, there was no protector in use. Typically, the situation was either unexpected or the protector had fallen off. In only 42 (3%) incidents had the conscript used a well-fitting hearing protector. In the remaining 142 (10%) cases there was a poor fit of the protector or another reason. In six questionnaires, no answer was given to this question.

### 3.4. Hearing Loss and Tinnitus

Tinnitus was the most prominent symptom, and the most typical hearing loss after AAT was seen at the frequency of 6 kHz. Tinnitus and other hearing symptoms were asked about separately from the conscripts in Group II. Tinnitus occurred after AAT in 201 (66%) out of the 306 subjects in this group, and other ear symptoms (pain, buzzing, ringing, whizzing, headache, balance-disturbance etc.) in 144 (47%). Both types of symptoms occurred in 125 (41%) among the 306 conscripts.

Noise caused damages for various group types of conscripts around the shot in 235 incidents. Most typically, two subjects (101 incidents), or 3–5 subjects (21–33 incidents depending on the group size) were exposed. Incidents with 6–12 subjects were encountered least frequently. For machine gun noise exposure, the most usual incident affected 2–6 conscripts (eight incidents). One example of an incident with a group of 27 conscripts without hearing protection was caused by six combat vehicles driving near the group and starting to shoot with machine guns with blank cartridges.

## 4. Discussion

We conducted a retrospective population-based 10-year study among all conscripts in the Finnish Defence Forces (FDF) during the years 1997–2003 and 2008–2010 to find out the frequency of acute acoustic trauma (AAT) after exposure to rifle-caliber noise exposure.

The main finding was that most AAT-related hearing losses occurred after unexpected firing of an assault rifle. Most incidents occurred at the distance of less than two meters, with a neighboring conscript shooting and because no hearing protectors were used. The distances were evaluated by the victims who claimed to have adverse effects. The distances were sometimes later certified by training officers. The AATs were usually not serious (10–20 dB), although maximum hearing losses varied between 50 to 80 dB.

The most prominent hearing loss after AAT in the present series typically affected the frequency band of 6 kHz. The same has been noted for the Finnish professional soldiers [9]. The exact comparison of hearing results before and after the AAT incidents is not possible, because the audiograms at the beginning of the service were measured by a screening audiometer only.

Although the hearing protectors are ordered to be used, the unexpected shots took place in circumstances where nobody in the training had hearing protectors. This forms a major challenge for the military training noise control. More importantly, it remains the responsibility of the training personnel to emphasize this even more in the rifle-handling of the conscripts. Unfortunately, false arguments still exist claiming that blank cartridges would not cause hearing loss. Developmental work to obtain blank cartridges with less noise emission from the muzzle of the rifle should be supported. There has been continuous product development in the muzzle brakes and wooden bullet breakers to enhance these efforts [16,17].

Communication hearing protectors are important in drill or shooting range activities, because there is always a risk of a shooting accident. There are many types of communication protectors when using earmuffs or plugs, and the variety is increasing.

Although technical orders for training sessions have been established according to the safety precautions, noise-related accidents still occur, and these can cause even permanent hearing deficits. It seems that the human control factors should be emphasized even in crisis simulation conditions. The impact of a hearing disorder on a conscript’s future career and communication skills is evident and should thus not be underestimated.

The present results are in accordance with earlier studies in Finland, Sweden and elsewhere [17,18,19,20]. For example, Muhr concluded that the incidence of decreased hearing thresholds was elevated during military service compared to the previous figures [13]. However, Holma pointed out that the risk of receiving a hearing loss during military service is variable [9].

Around 0.7–1.5% of the conscripts experience an AAT during their service in the FDF. Vanmaele et al. [21] studied military AAT-related hearing losses in Belgium and found a prevalence of 0.6%. In the French armed forces, the AATs decreased during the period from 2007 to 2014. The AAT incidence rate varied between 4.0 cases per 1000 person-years in 2011 and 3.4 in 2013 [22]. In the Republic of Korea 2.8% of all military personnel reported a hearing loss [23], and the use of hearing protection was modest. The present data regarding the FDF are in accordance with these studies. Further, in an earlier questionnaire study involving 416 conscripts in the FDF, personal experience of tinnitus was related both to longer service time and high levels of noise exposure [4].

Our study includes certain limitations. Audiometric analysis results can occasionally be dependent on the attitude of conscripts at the end of their military service period as some individuals may seek compensation for their AAT-induced hearing loss. On the other hand, it is also possible that some conscripts did not complain of their hearing symptoms after noise exposure, and thus the damage was not registered. The amount of missing data was a minor factor in the present study. Furthermore, the possibility of combined effects on the background of the registered AAT, i.e., simultaneous heavy weapon shooting, blows on the ear, leisure time noise exposure, etc., remains. Our results on AAT-induced hearing loss and tinnitus are impossible to verify using a control group. The present series comprised 1674 of a cohort of 220,000 conscripts. In Finland currently about 70–80% of the male population undergo military training, and therefore it is difficult to obtain a control group for comparison. Furthermore, those with an earlier permanent sensorineural hearing loss or other major health problems get an exemption from military service.

## 5. Conclusions

We conclude that around 0.7–1.5% of conscripts experience an AAT during their service with the Finnish Defence Forces. Most incidents occur after an accidental shot with an assault rifle, when persons around the weapon do not use hearing protection. Firing a blank cartridge as an accidental shot seems to be most typical at these occasions. Faults in weapon handling and carelessness regarding hearing protection in the battlefield military training form the basis for these. Training, control measures, and raising the awareness of possible consequences of ATT are the most important possibilities to reduce the incidence of these accidents.

## Figures and Tables

**Figure 1 ijerph-20-03366-f001:**
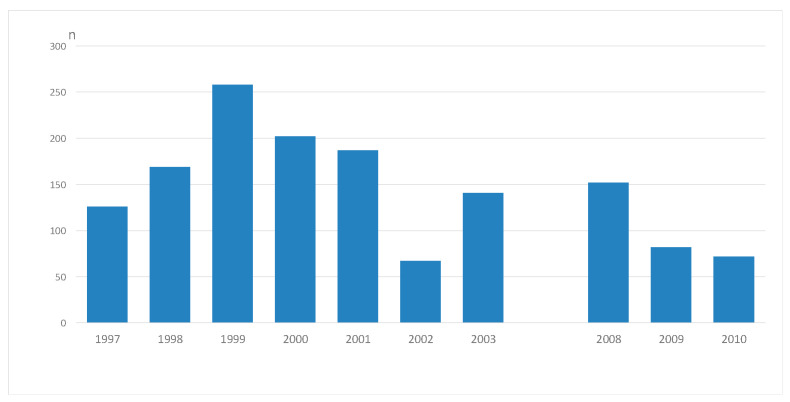
Annual number (n) of conscripts with a new hearing loss after an exposure to assault-rifle noise during 1997–2003 and 2008–2010.

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
