# Peer review of "Acute Acoustic Trauma after Exposure to Assault Rifle Noise among Conscripts in the Finnish Defence Forces—A Population-Based Survey"

_ijerph, 2023, doi:10.3390/ijerph20043366_

Round 1
Reviewer 1 Report
Dear Colleagues.
You have studied an extensive number of subjects, congratulations.
Can you explain the very high number of accidents in 1999?
There are always years with very small numbers of accidents. How do you explain this? For example, 2009 and 2010.
Hyperbaric oxygen therapy is given as the therapy. What is your position on glucocorticoid therapy? Is this used in such cases?
In how many patients does restitution occur?
If acoustic trauma occurs in 0.7-1.5% of cases, why is a qualified hearing test not performed prior to military service?
The literature cited includes primarily Finnish authors. What is the situation in other countries of the world (USA, UK, Germany)?
Author Response
Letter of Responses
We would like to thank the Reviewer for the excellent comments and suggestions that have certainly improved our manuscript. Our responses Arte listed below and the changes can be found in the revised manuscript.
REVIEWER 1
Can you explain the very high number of accidents in 1999? There are always years with very small numbers of accidents. How do you explain this? For example, 2009 and 2010.
RESPONSE: We agree with the Reviewer that this observation is interesting. We checked the annual numbers and the only explanation that remains obvious for this issue is that there are different types of accidents causing ATTs. During some years, and in certain accidents, groups of conscripts vs. sometimes only a single person experienced ATT at certain incidents. This has consequently had an impact on the annual numbers.
Hyperbaric oxygen therapy is given as the therapy. What is your position on glucocorticoid therapy? Is this used in such cases?
RESPONSE: This is a valid point and we have now added the following two sentences in the revised manuscript (page 3, paragraph 3): Currently, corticosteroids and hyperbaric oxygen therapy are the standard treatment of acoustic trauma in the FDF. However, during the study periods, normobaric or hyperbaric oxygen only was used.
In how many patients does restitution occur?
RESPONSE: This is of course an interesting issue, but as we have not been able to investigate this matter in the present series, we did not add any information to our manuscript. However, the State Treasury is the agency in charge of accident compensation payments to conscripts and military personnel. The decision of compensation is always individual. We found one Finnish official report stating that compensation by the Finnish state (the State Treasury) for conscripts having experienced an ATT was paid in 227 cases during 2011-2015.
If acoustic trauma occurs in 0.7-1.5% of cases, why is a qualified hearing test not performed prior to military service?
RESPONSE: This is an important issue. Prior to military service all Finnish recruits are examined (screened) for hearing. The quality of these hearing screening tests has proved to be of good quality (Kokkonen J, Varonen S. Reliability of Primary Health Care Audiograms by Non-qualified Examiners-An Analysis of 1,224 Cases Otol Neurotol 2021 1;42(3):e261-e266.). We have now added this information and reference in the Introduction part (page 2, paragraph 6).
The literature cited includes primarily Finnish authors. What is the situation in other countries of the world (USA, UK, Germany)?
RESPONSE: We have now added these international reports as new references to the revised version of our manuscript:
Shooting habits of youth recreational firearm users.
Stewart M, Meinke DK, Snyders JK, Howerton K.Int J Audiol. 2014 Mar;53 Suppl 2:S26-34. doi: 10.3109/14992027.2013.857437.PMID: 24564690
Noise Exposure on a Warship during Firing of a Heavy Machine Gun.
Paddan GS. Ann Occup Hyg. 2015 Nov;59(9):1208-11. doi: 10.1093/annhyg/mev053. Epub 2015 Aug 2.PMID: 26240198
Occupational noise exposure on a Royal Navy warship during weapon fire.
Paddan GS. Noise Health. 2016 Sep-Oct;18(84):266-273. doi: 10.4103/1463-1741.192474.PMID: 27762256 Free PMC article.
Reviewer 2 Report
The object matter of this study could be interesting for the readers if the authors describe the acoustic trauma in more detail from a pathophysiological point of view, define the study group more precisely and express concrete results based on a rather statistical design than making a meaningless list of percentages. In my opinion, the whole discussion loses its meaning if there are no statistical calculations that express the significance of the results obtained in this study group. I therefore strongly suggest a review by a professional statistician or one of the authors who possesses these skills.
ABSTRACT
Line 12 It would be better to write “the aim of the present study was to…”
Line 13 This nationwide population-based cohort comprised all conscripts (>220 000) …can we know the exact number of conscripts in the population analyzed?
Line 21 Please, include “in conclusion” in your phrase.
INTRODUCTION
It is a retrospective review of noise damage cases, but the authors get lost in the description of the military organization in Finland and hearing protectors, without describing the noise damage at all. I recommend to better describe what acute noise damage is, in particular differentiating hearing fatigue (the temporary raising of the hearing threshold) which I think this article is talking about, from the permanent hearing loss which consists of chronic noise damage. it would also be appropriate to recall the pathophysiological mechanisms that lead to cochlear damage as a result of exposure to noise.
In fact, the introduction does not provide a clear definition of noise-induced cochlear injury; the authors (as I write for the part of methods) should choose a clear definition of noise-induced hearing loss to make the study group homogeneous. Furthermore, it is serious that they never mention in the text the “sensorineural” hearing loss. Everything should be more precise from an audiological point of view.
MATERIALS AND METHODS
Lines 80-90: Authors are not very clear about the inclusion and exclusion criteria of their study cohort. In the first lines they say to have included all conscripts with symptoms of AAT (ear pain, tinnitus, hearing deficit…) but in the following lines they describe a minimum hearing deficit of 10db or 15db outside the speech area.
Authors should better explain all the inclusion criteria for their study, alternatively this description should be included in the results of the study.
Line 88: Why authors change the inclusion criteria (or their definition of AAT) from 2006? What recommendations do they refer to in the text? Please insert reference.
I recommend making the inclusion criteria of the study group homogeneous regardless of the reference year and using a clear definition of acoustic noise trauma. Authors should state if they included only people with hearing loss (HL) or also people with symptoms (such as tinnitus) without hearing loss. In the first case, a clear definition of HL for acute acoustic trauma should be adopted; in the second case, a differentiation between the group with symptoms without HL and the group with HL should be done.
It seems that they use AAT and hearing loss as synonyms but it is not correct.
RESULTS
In the abstract authors write “We found that 0.7-1.5% of the conscripts experienced an AAT during their 21 service in the FDF”. Does “AAT” stand for hearing loss from AAT? Does this percentage derive from the 1617 cases of HL out of what total number? Please state this information in the abstract and in the results line 123.
Line 125: Please replace the term “hearing losses” with “cases of HL”.
Line 133: “There were 1439 (99%) males in the study cohort and 17 females (1%)” ...is the total number 1456? Why don’t you consider 1617 as the total number of cases of HL, rather than considering only those from rifle caliber weapons? Can we know something more about the group other than the sex (mean age and range for example)?
Line 137: Is the diminishing trend significative? Why don’t you consult a statistician? Considering the total case of HL, is the use of a rifle caliber weapon a risk factor for an acoustic trauma? I think that the help of a statistician should improve the meaning of this study.
Line 147: 1286/…, 89%. For each percentage it should be better to specify the total number.
Line 155: What was the number of cases of hearing loss although well-fitted hearing protectors? 42 or 37? It is not clear.
Line 159: “No significant difference was noted in the use of hearing protectors between Group I and II (data not shown)”. Do you mean statistically significant? Please explain otherwise cancel.
Figure 2: Please insert the meaning of the two axes in the figure.
How can you consider a mean HL less than 10dBHL if in the methods you state to have considered only HL > 10dBHL?
“Averages of hearing thresholds (n=542) and maximum hearing threshold”. Do you mean hearing thresholds shift?
I don’t understand the figure of maximum hearing threshold…please explain.
Probably the graphs should be divided into two different figures, otherwise it would be even better to replace everything with a boxplot with median, mean, minimum and maximum value for each frequency.
Table 1: the table thus organized with these average values ​​has no meaning. Rather than indicating the class mean value, rearrange the table by indicating the percentage of cases of HL for each class and the percentage of people with tinnitus overall and by hearing loss class. For example, it would be interesting to know whether the patients with the highest class developed tinnitus more.
Furthermore, how can you count cases with a hearing threshold lower than 20dB if you say you have excluded them in the methods? everything is unclear and very imprecise.
Author Response
We would like to thank the Reviewer for the excellent comments and suggestions that have certainly improved our manuscript. Our responses Arte listed below and the changes can be found in the revised manuscript.
REVIEWER 2
The object matter of this study could be interesting for the readers if the authors describe the acoustic trauma in more detail from a pathophysiological point of view, define the study group more precisely and express concrete results based on a rather statistical design than making a meaningless list of percentages.
RESPONSE: We have now added the following description regarding the pathophysiology of an ATT in the Introduction part (page 2, paragraph 2):
A strong acoustic trauma will induce histological changes in the cochlea and auditory nerve. Cochlear outer hair cell loss and damage of inner hair cell stereocilia and axonal loss and myelin sheath disorganization of the auditory nerve will consequently lead to permanent hearing loss. (Coyat C, Cazevieille C, Baudoux V, Larroze-Chicot P, Caumes B, Gonzalez-Gonzalez S. Morphological consequences of acoustic trauma on cochlear hair cells and the auditory nerve. Int J Neurosci. 2019 Jun;129(6):580-587.)
In my opinion, the whole discussion loses its meaning if there are no statistical calculations that express the significance of the results obtained in this study group. I therefore strongly suggest a review by a professional statistician or one of the authors who possesses these skills.
RESPONSE: We agree with the Reviewer that the significance of the present results would warrant statistical proof. However, the annual numbers vary greatly i.e. between 50-250 per year, and this is obviously due to groups of conscripts having been exposed to fire-arm noise at certain accidents. It thus remains challenging and potentially misleading to draw statistical conclusions by comparing these variable numbers.
ABSTRACT
Line 12 It would be better to write “the aim of the present study was to…”
RESPONSE: This has now been corrected accordingly.
Line 13 This nationwide population-based cohort comprised all conscripts (>220 000) …can we know the exact number of conscripts in the population analyzed?
RESPONSE: Unfortunately, we are not allowed to report the exact numbers of conscripts in the FDF. We have used the available annual numbers of conscripts to obtain the sum of these and then only use the figure ‘>220 000’. Hopefully, this will be feasible in this paper.
Line 21 Please, include “in conclusion” in your phrase.
RESPONSE: This has now been corrected accordingly.
INTRODUCTION
It is a retrospective review of noise damage cases, but the authors get lost in the description of the military organization in Finland and hearing protectors, without describing the noise damage at all. I recommend to better describe what acute noise damage is, in particular differentiating hearing fatigue (the temporary raising of the hearing threshold) which I think this article is talking about, from the permanent hearing loss which consists of chronic noise damage. it would also be appropriate to recall the pathophysiological mechanisms that lead to cochlear damage as a result of exposure to noise.
In fact, the introduction does not provide a clear definition of noise-induced cochlear injury; the authors (as I write for the part of methods) should choose a clear definition of noise-induced hearing loss to make the study group homogeneous. Furthermore, it is serious that they never mention in the text the “sensorineural” hearing loss. Everything should be more precise from an audiological point of view.
RESPONSE:
We agree with the Reviewer and have now defined and expressed these terms more accurately. We now use the term ‘sensorineural hearing loss’ to differentiate the condition from any conductive disorders.
Of note, about 15-20% of young males have an elevated audiometric threshold shift before they enter military service, which must be separated from the hearing loss received during military service.
MATERIALS AND METHODS
Lines 80-90: Authors are not very clear about the inclusion and exclusion criteria of their study cohort. In the first lines they say to have included all conscripts with symptoms of AAT (ear pain, tinnitus, hearing deficit…) but in the following lines they describe a minimum hearing deficit of 10db or 15db outside the speech area.
RESPONSE:
We have now modified the text as follows to be more accurate (page 2, paragraph 7).
The inclusion criterium was that if the conscript complained changed hearing after exposure to assault rifle noise, then a questionnaire was filled and later an audiometric analysis was performed. All conscripts having filled the questionnaire were thus included in this study. It was still possible that the audiometric analysis later showed only a rather limited hearing threshold shift.
Authors should better explain all the inclusion criteria for their study, alternatively this description should be included in the results of the study.
RESPONSE: Please, see our previous response.
Line 88: Why authors change the inclusion criteria (or their definition of AAT) from 2006? What recommendations do they refer to in the text? Please insert reference.
RESPONSE: The inclusion criteria were the same for both study Groups I and II, but the frequency content of audiometric analyses was only available and analyzed for Group II.
I recommend making the inclusion criteria of the study group homogeneous regardless of the reference year and using a clear definition of acoustic noise trauma. Authors should state if they included only people with hearing loss (HL) or also people with symptoms (such as tinnitus) without hearing loss. In the first case, a clear definition of HL for acute acoustic trauma should be adopted; in the second case, a differentiation between the group with symptoms without HL and the group with HL should be done.
RESPONSE:
We agree with the Reviewer about the chance of misunderstanding in this issue and we have thus expressed this now more clearly in the text (page 2, paragraph 7). The inclusion criteria were the same for both groups (Group I and II) but the type of data were different.
It seems that they use AAT and hearing loss as synonyms but it is not correct.
RESPONSE: We agree with the Reviewer, these terms are not synonyms. AAT can lead to hearing loss and hearing loss can be due to various other etiologies. We have now corrected this issue in the revised manuscript.
RESULTS
In the abstract authors write “We found that 0.7-1.5% of the conscripts experienced an AAT during their 21 service in the FDF”. Does “AAT” stand for hearing loss from AAT? Does this percentage derive from the 1617 cases of HL out of what total number? Please state this information in the abstract and in the results line 123.
RESPONSE:
This is an important point of our study. The calculation is based on the figures 1617/220 000 x 100% = 0,7 %. However, in practice there are slightly less conscripts who have been exposed to long lasting intensive impulse noise (assault rifle). Further, usually all conscripts visit a shooting range at least three times during their military service. So, every conscript will be exposed to at least 100 cartridges and in addition to the noise exposure during military drills. Therefore, we have estimated the figure to be between 0.7-1.5%.
Line 125: Please replace the term “hearing losses” with “cases of HL”.
RESPONSE: This has been corrected accordingly (“cases of HL from rifle caliber noise”).
Line 133: “There were 1439 (99%) males in the study cohort and 17 females (1%)” ...is the total number 1456? Why don’t you consider 1617 as the total number of cases of HL, rather than considering only those from rifle caliber weapons? Can we know something more about the group other than the sex (mean age and range for example)?
RESPONSE: These numbers are explained by the fact that part of the hearing losses were caused by heavy weapons (not rifle caliber) or explosions, and these will be analyzed separately and are not included in the present study.
Line 137: Is the diminishing trend significative? Why don’t you consult a statistician? Considering the total case of HL, is the use of a rifle caliber weapon a risk factor for an acoustic trauma? I think that the help of a statistician should improve the meaning of this study.
RESPONSE: We agree with the Reviewer that the significance of the present results would warrant statistical proof. However, the annual numbers vary greatly i.e. between 50-250 obviously due to groups of conscripts having been exposed to fire-arm noise at certain accidents. It thus remains challenging and potentially misleading to draw statistical conclusions by comparing these variable numbers.
Line 147: 1286/…, 89%. For each percentage it should be better to specify the total number.
RESPONSE: We have now corrected this accordingly.
Line 155: What was the number of cases of hearing loss although well-fitted hearing protectors? 42 or 37? It is not clear.
RESPONSE: The correct number is 42 and to make this clearer, we have now omitted the sentence with the number 37 describing those incidents where the distance to the rifle was less than 2 meters.
Line 159: “No significant difference was noted in the use of hearing protectors between Group I and II (data not shown)”. Do you mean statistically significant? Please explain otherwise cancel.
RESPONSE: We agree with the Reviewer that this sentence lacks the required accuracy and we have now omitted the sentence
Figure 2: Please insert the meaning of the two axes in the figure. How can you consider a mean HL less than 10dBHL if in the methods you state to have considered only HL > 10dBHL? “Averages of hearing thresholds (n=542) and maximum hearing threshold”. Do you mean hearing thresholds shift? I don’t understand the figure of maximum hearing threshold…please explain. Probably the graphs should be divided into two different figures, otherwise it would be even better to replace everything with a boxplot with median, mean, minimum and maximum value for each frequency.
RESPONSE: We agree with the Reviewer that the information in this Figure is hard to understand as it describes the highest threshold shift for an individual person. We have now omitted the Figure from the revised version of the manuscript.
Table 1: the table thus organized with these average values ​​has no meaning. Rather than indicating the class mean value, rearrange the table by indicating the percentage of cases of HL for each class and the percentage of people with tinnitus overall and by hearing loss class. For example, it would be interesting to know whether the patients with the highest class developed tinnitus more.
Furthermore, how can you count cases with a hearing threshold lower than 20dB if you say you have excluded them in the methods? everything is unclear and very imprecise.
RESPONSE: We agree with the Reviewer that the information from Table 1 remains too limited and therefore we omitted the Table from the revised version of our manuscript.
Round 2
Reviewer 1 Report
Dear authors!
I congratulate you on the now very good amended version.
I have no further points of criticism to add.
With best regards
Author Response
We thank the Reviewer for all efforts and the kind comments regarding our manuscript.
Reviewer 2 Report
The authors improved the quality of the manuscript in particular by clarifying the group inclusion and exclusion criteria and the definition of hearing impairment.
I am sorry that they hastily decided to delete figure 2 and table 1 rather than improve their quality. I still believe that the help of a statesman could improve the meaning of the study.
Figure 2: Please insert the meaning of the two axes in the figure.
How can you consider a mean HL less than 10dBHL if in the methods you state to have considered only HL > 10dBHL?
“Averages of hearing thresholds (n=542) and maximum hearing threshold”. Do you mean hearing thresholds shift?
I don’t understand the figure of maximum hearing threshold…please explain.
Probably the graphs should be divided into two different figures, otherwise it would be even better to replace everything with a boxplot with median, mean, minimum and maximum value for each frequency.
Table 1: the table thus organized with these average values ​​has no meaning. Rather than indicating the class mean value, rearrange the table by indicating the percentage of cases of HL for each class and the percentage of people with tinnitus overall and by hearing loss class. For example, it would be interesting to know whether the patients with the highest class developed tinnitus more.
Furthermore, how can you count cases with a hearing threshold lower than 20dB if you say you have excluded them in the methods? everything is unclear and very imprecise.
Author Response
Response Letter for the Reviewer 2
The authors improved the quality of the manuscript in particular by clarifying the group inclusion and exclusion criteria and the definition of hearing impairment.
I am sorry that they hastily decided to delete figure 2 and table 1 rather than improve their quality. I still believe that the help of a statesman could improve the meaning of the study.
Figure 2: Please insert the meaning of the two axes in the figure.
How can you consider a mean HL less than 10dBHL if in the methods you state to have considered only HL > 10dBHL?
“Averages of hearing thresholds (n=542) and maximum hearing threshold”. Do you mean hearing thresholds shift?
I don’t understand the figure of maximum hearing threshold…please explain.
Probably the graphs should be divided into two different figures, otherwise it would be even better to replace everything with a boxplot with median, mean, minimum and maximum value for each frequency.
RESPONSE: Thank you for your considerations regarding our manuscript and the received comments. We are grateful for the previous criticism regarding Figure 2 and thus agreed to omit the figure. In terms of clarity, the main message of our manuscript is more obvious and understandable without the Figure 2. Further, it is our firm opinion, that statistical analysis would not improve the main message due to the limited and heterogenous data in terms of variation between the included years.
Table 1: the table thus organized with these average values has no meaning. Rather than indicating the class mean value, rearrange the table by indicating the percentage of cases of HL for each class and the percentage of people with tinnitus overall and by hearing loss class. For example, it would be interesting to know whether the patients with the highest class developed tinnitus more.
Furthermore, how can you count cases with a hearing threshold lower than 20dB if you say you have excluded them in the methods? everything is unclear and very imprecise.
RESPONSE: Thank you for the comment. We agree that in present form, the average values have no meaning. We would thus like to omit this table as its information remains limited. Our data are still relatively modest in terms of its volume and generalizability, and there is better information available on military hearing loss, hearing loss class and tinnitus (see for example: Mrena R. INVESTIGATIONS OF NOISE-RELATED TINNITUS. Department of Otorhinolaryngology – Head and Neck Surgery. Academic Thesis. University of Helsinki, Helsinki, Finland 2011. https://helda.helsinki.fi/bitstream/handle/10138/27440/investig.pdf?sequence=1&isAllowed=y)
However, inspired by the suggestion of the Reviewer, we are now aiming at gathering more data on a larger series on military noise. This future research proposal will cover a broader concept than adverse effects among conscripts only.
To add text about the correlation of tinnitus and military noise exposure we have now added the following sentence to the Discussion part (row 250) and we refer to the reference (4).
“In an earlier questionnaire study on 416 conscripts in the FDF, personal experience of tinnitus was related both to longer service time and high levels of noise exposure”.
(4) Jokitulppo J, Toivonen M, Pääkkönen R, Savolainen S, Björk E, Lehtomäki K. Military and leisure-time noise exposure and hearing thresholds of Finnish conscripts. Mil Med 2008 Sep;173(9):906-12. doi: 10.7205/milmed.173.9.906.
